# Harnessing the Power of Eph/ephrin Biosemiotics for Theranostic Applications

**DOI:** 10.3390/ph13060112

**Published:** 2020-06-01

**Authors:** Robert M. Hughes, Jitka A.I. Virag

**Affiliations:** 1Department of Chemistry, East Carolina University, Science and Technology Building, SZ564, East 5th St, Greenville, NC 27858, USA; hughesr16@ecu.edu; 2Department of Physiology, Brody School of Medicine, East Carolina University, Warren Life Sciences Building, LSB239, 600 Moye Blvd, Greenville, NC 27834, USA

**Keywords:** Eph/ephrin, receptor tyrosine kinase, therapeutic target, smart drug, nanoparticles, liposomes, exosomes

## Abstract

Comprehensive basic biological knowledge of the Eph/ephrin system in the physiologic setting is needed to facilitate an understanding of its role and the effects of pathological processes on its activity, thereby paving the way for development of prospective therapeutic targets. To this end, this review briefly addresses what is currently known and being investigated in order to highlight the gaps and possible avenues for further investigation to capitalize on their diverse potential.

## 1. Introduction

Detection, transduction and appropriate response behavior of neighboring cells are essential components of optimal communication and the coordinated and integrated function necessary for survival. In a complex system, this is determined by the summative and timely intracellular processing of a myriad of intersecting signals from various distances and directions. The Eph receptors and ephrin ligands are the largest tyrosine kinase (RTK) superfamily, comprised of 22 known members, subdivided into A- and B-subclasses, and are heterogeneously expressed in almost every tissue [1]. These membrane-anchored proteins exhibit unique cell-to-cell communication via bidirectional signaling, which modulates cytoskeletal dynamics and thus cell–cell recognition and motility, making them integral mediators in developmental processes and the maintenance of homeostasis [2,3]. Given the importance of these biomolecules, it stands to reason that derangements in expression and signaling sequelae would contribute to and/or cause disease [4,5,6,7,8,9,10]. The disparate influences of aging, ethnicity, and gender on expression and signaling variations, however, which may exacerbate pre-existing imbalances, remain largely unexplored. Their complicated expression profiles, often with multiple ligands and/or receptors on one cell adjacent to or near other(s) with a different expression profile, compounded by their various affinities for receptor binding, oligomerization, cis- and trans-activation or inhibition, all contribute to the complicated delineation of their roles in normal physiology and disease progression, as well as development of customized, targeted therapeutics [11,12,13,14,15,16,17].

## 2. Eph/ephrin is a Ubiquitous Therapeutic Target

The Eph receptors (Eph = erythropoietin-producing hepatocellular) and their congeneric ligands, the ephrins (contraction of “Eph receptor interacting proteins” and after the Greek word “ephoros” meaning overseer or controller), are the largest family of receptor tyrosine kinases, comprised of 14 receptors and 8 ligands. Since their discovery and cloning by Hirai et al., published in Science in 1987 [18], more than 5000 articles have reported on the structure of these proteins, cellular and tissue expression profiles, downstream signaling mechanisms, and their contribution to the differentiation, proliferation and migration of various cell types. Ligands A1–A5 and B1–B3 are typically membrane-anchored, and cell-to-cell contact is usually but not always required, resulting in binding with varying affinity and promiscuity to the receptors A1–A8, A10, B1–B4, and B6. Subsequent dimerization, tetramerization and/or clustering of ligands and receptors (Figure 1A) can cause activation or silencing of bi-directional signaling and consequent activation (or inhibition) of intracellular cascades in each cell [19,20,21,22,23,24]. Ojosnegros et al. (2017) recently adapted enhanced fluorescence fluctuation imaging analysis to resolve the spatial relationships of polymerization and formation of larger aggregates, generating a model of polymerization-condensation dynamics which suggests that these associations result not only in amplification of the signal, but also in termination [25]. To complicate matters, domain mutation analyses have revealed that the nature of the contact interfaces in the ligand-binding domains confer binding specificity, but it is the domains contained within ectodomains (e.g., SAM—sterile alpha motif) that are indispensable for localization and subsequent forward intracellular signaling [26,27,28]. Lastly, the importance of the lipid bilayer composition [29], as well as its interface as a regulator of their orientation and configuration with respect to the membrane, as demonstrated by charge-swapping simulations [30], illustrates their effectual plasticity.

The multifaceted and ubiquitous expression of Eph/ephrin RTKs in nearly all cells of the body, although most extensively studied in cancer and development, implicates them in the majority of vital physiologic processes (Figure 1B) [3]. The knowledge we have acquired from these studies has generated valuable information about the role of Eph/ephrin RTKs in cellular communication and behavior, providing a wealth of therapeutic targeting opportunities that may be exploited for a range of pathologies, including but not limited to atherosclerosis [5,31,32], neurodegenerative/cognitive and endocrine disorders [33,34,35,36,37,38], gastrointestinal and genito-urinary maladies [39,40,41], immune disorders [4,5,9], musculoskeletal growth and metabolism diseases [42,43,44,45], ischemic tissue injury [46,47,48,49,50,51,52,53,54,55,56] and malignancies [7,57,58], reproductive/fertility illnesses [59,60,61,62] and organ fibrosis [10,63]. A recently observed connection between viral infections and tumorigenesis mediated through Eph/ephrins is also being characterized, raising the possibility of alternative anti-viral strategies [64]. This review aims to illustrate the potential for developing targeted vehicles, targeting methods, and therapeutic cargo to modulate Eph/ephrin signaling (Figure 2).

## 3. Eph/ephrin Signaling Dynamics

Eph receptors and ephrin ligands are expressed in a variety of combinations and permutations in several cell types. These multi-domain proteins each contain several regions: the extracellular ligand-binding domain, a cysteine-rich domain, two fibronectin domains, FN1 and FN2, the cytoplasmic kinase domain, and the SAM domain [11,12,17,65]. These, in conjunction with aforementioned structural features, downstream effectors, termination by ADAMs (a disintegrin and metalloprotease), and recently discovered non-catalytic and reciprocal cross-regulatory activities, represent a treasure trove of targeting possibilities [13,15]. The following paragraphs discuss some of what is known about the role of Eph/ephrin signaling merely as an introduction to some common aspects of cellular function. Specifically, cell repulsion/adhesion, migration and proliferation, as drivers of inflammation as well as vascular and lymphatic function, are features of cells in nearly all organs (Figure 1B; [6,66]), and have broad applications in most disease processes.

Attenuation of inflammation and/or vascular permeability via Eph/ephrin signaling modulation significantly reduces tissue injury [5,9,41,67]. In the vascular endothelium and leukocytes, Eph/ephrin expression can be enhanced by pro-inflammatory cytokines, and can regulate local as well as systemic immune cell trafficking [4,9,68,69]. Ephrin-B2 and EphB4 are known to be expressed on the vascular endothelium and play a role in hematopoiesis and cell mobilization [70]. Luminal ephrinB1 and B2 expression are upregulated in endothelial cells, and interact with EphB2 on macrophages to promote transmigration [71]. Endothelial EphA2 is upregulated in response to TNF-α [72], and its activation enhances proinflammatory NF-κB and promotes disruption of the endothelial/epithelial barrier [5]. There is mounting evidence that both Eph receptors and ephrin ligands may also influence toll-like receptor function to propagate immune and/or tumorigenic signaling pathways, but more information about their prevalence and changes as an effect of disease on innate immune cells is needed [73]. EphA2, ephrin-A1 and ephrin-A2 are upregulated during monocyte differentiation [74] and, as inflammation progresses, leukocytic EphA1, EphA3 and EphA4 promote clustering of adhesion molecules and permit selective infiltration, resulting in adherence to endothelial cells and contributing to disruption of vascular integrity [5,68]. Cytoskeletal reorganization required for migration of T lymphocytes, widely recognized for their role in immune surveillance and initiation of the immune response, may be influenced by EphA/ephrin-A1 signaling, since T cells express Eph A1, A2, A3, A4, A7 and A8 [75,76]. Ephrin-A1 stimulation of T cells prevents chemotaxis, and thus ephrin-A1 may be a negative regulator of inflammation [77]. Sharfe et al. (2008) propose that ephrin-A1 enhances integrin-mediated adhesive interactions between T cells and endothelial cells, while EphA receptor activation on T cells inhibits it, so this would restrict T cell transmigration. Members of the EphB/ephrin-B family, integral mediators of vascular, lymphatic and neuronal development [78,79,80], also play a role in many of these disease processes, as observed by the involvement of EphB1 in microglial cells in models of nociceptive pain and inflammation [81], as well as splenic mononuclear EphB4-mediated inflammatory bowel disease [41]. Of course, expression of A and B receptors and ligands in any given cell within a tissue are not mutually exclusive, nor does their presence preclude the possibility for changes, in expression level and/or isoform, when that type of cell is detected in other tissue beds and/or exposed to various stimuli. For example, B cells in normal human peripheral blood express ephrinA4, EphA1-A4, EphA8, EphB2 and EphB4 while lymph node samples contain different amounts of ephrin-A4 and EphA1-A4 as well, but also ephrin-A1, ephrin-A3, ephrin-A5, EphA5, EphA10, ephrin-B2, EphB1 and EphB6; expression profiles that change significantly with disease progression [82]. T cells have also been shown to express EphB1, EphB2, EphB3, EphB6 and ephrinB1-B3, but whether there is co-expression on the same T cell and how they contribute to activation and/or differentiation are currently under investigation [9]. Adaptor proteins, a large group of accessory proteins containing protein-bind motifs, often have multiple binding units that enable them to join other proteins together to create a larger signaling complex. In particular, Src-like adaptor proteins (SLAPs), expressed primarily in the immune system, are involved in Eph RTK-mediated signal transduction by virtue of their interaction with several other adaptor/effector proteins, and the resultant activation or inhibition differentially influences tumorigenesis in a variety of tissues [83].

The role of Eph/ephrin signaling in the development of the lymphatic system has long been established. In contrast to the exclusive expression of ephrin-B2 and EphB4 in arteries and veins, respectively, an essential feature that drives boundary formation of these closely opposed systems [84], both ligand and receptor are expressed on lymphatic vessels [85]. Importantly, the PDZ domain (an initialism representing a common structural domain in signaling proteins that serves as an interface between the membrane and the cytoskeleton), but not the kinase domain, of the ephrin-B2 ligand is required for reverse signaling to promote normal lymphangiogenesis during development [86]. It is known that the lymphatic system, normally quiescent in an adult, is the biological vehicle for malignant metastasis [58] and, given the prominent role of Eph/ephrin RTKs in tumorigenesis and metastasis, their suitability as molecular targets is an active area of investigation that holds great promise [87]. Indeed, the potential utilization of Eph/ephrin signaling as prognostic indicators of disease progression or therapeutic targets, as in the correlation of EphB4/ephrin-B2 levels with the incidence of metastasis in breast cancer survivors [88], and the efficacy of targeting ephrin-B2 to limit tumor growth via attenuating adverse lymphangiogenesis and angiogenesis [89], are increasingly being recognized.

The importance of Eph/ephrin signaling during development underscores their relevance to stem cell plasticity, and they have been reported as determinants of cell fate in embryonic, mesenchymal and cord blood stem cells. As previously discussed, Eph/ephrin signaling participates in hematopoiesis and related malignancies. Mesenchymal stems from bone marrow and adipose tissue express several members of both the A and B families, which regulate survival and renewal, differentiation, morphology, mobilization, homing and engraftment [90,91,92]. In embryonic stem cells, a phosphoproteomic study revealed that the EphA2 receptor was reported to be a critical target of FGF4, leading to disabling of the EphA2 receptor via ERK1/2 and ser/thr phosphorylation and concomitant transcription of ephrin ligands, resulting in exit from the pluripotent compartment to initiate differentiation [93]. Reprogramming of adult somatic stem cells is effected by secreted, truncated EphA7-induced attenuation of ERK1/2 signaling [94]. Oleic acid, a naturally occurring mono-unsaturated fatty acid, increased EphB2 expression in cord blood-derived mesenchymal stem cells, thereby enhancing their migratory capacity by increasing F-actin formation which alters the cytoskeleton [95]. Several members of the Eph/ephrin RTK family contribute to blastocyst implantation and invasion, as well as modulation of immunity during placentation [96]. The complex nature of Eph/ephrin signaling in these stem cell populations, as a function of space and time, warrants systematic characterization in order to fully appreciate their role in development and disease.

The Eph/ephrin RTKs clearly exert a prominent influence on cytoskeletal reorganization, which regulates cellular adhesion/repulsion/tensity, key elements of morphologic changes in metastasis and pathologic angiogenesis, [97,98] as well as modulation of inflammatory cell motility [99,100]. Reverse signaling through ephrinB1, resulting in tyrosine phosphorylation, recruits SH2/SH3 domain adaptor protein Grb4 binds and subsequently activates FAK, Cbl, Abl, paxillin and others to modulate cytoskeletal dynamics [101,102]. Forward signaling via autophosphorylated EphB2 and concomitant recruitment of myosin 1b regulates cytoskeletal remodeling [102,103]. EphA1/Ephrin-A1 activation attenuates RhoA/ROCK (Rho-associated coiled-coil containing protein kinase) to modify the cytoskeleton, and thus influences morphology and motility in HEK293 cells [104]. Shp2 regulates dephosphorylation of ROCKII, and adds an additional level of control to RhoA-dependent activation and mediation of contractile forces [105], and Shp2 is recruited by ephrin-A1-EphA2 leading to suppression of integrin function and FAK inactivation, which has strong implications for inflammation and wound-healing processes [106]. Other aspects of wound-healing, such as Ca2+ handling [107], mitochondrial fission [108], ATP generation [109], apoptosis [8,110], autophagy and fibrosis [111], are also influenced by this system, and there may well be others as yet undiscovered. Intracellular post-translational signaling cascades and transcriptional modifications can also be affected, but the mediators involved in transmitting these signals are, as yet, unclear. To complicate matters, cancer progression associated with an Eph receptor can be paradoxically up- or down-regulated in a tissue-specific manner [7]. Moreover, post-translational modifications can govern subcellular localization, conformation and clustering, which will directly influence activity, a feature which may be manipulated for therapeutic advantage. For example, the pharmacodynamics of EphA4-Fc (a chimeric protein fused with the Fc constant region of immunoglobulin heavy-chain) were substantially enhanced by glycosylation [112]. Scheideler et al. (2020) have developed a novel high-throughput DNA-based patterning platform, using multilayer lithography to recreate the complex spatial signaling systems between cells and tissues, that will accelerate our understanding of how these signals cumulatively govern cell behavior [113]. Much more research is needed to delineate the diverse roles of this system in a cell- and tissue-specific manner, as well as its systemic interplay under normal as well as pathological conditions.

## 4. Small Molecule, Peptide, Protein and RNA Targeting of Ephrins

Advancements in the automated manufacturing and screening of small molecules with increased potency and specificity are anticipated to accelerate the identification of viable targets and yield promising therapeutic candidates. While screening of small molecule libraries remains a productive strategy for the identification of novel Eph agonists and antagonists, design strategies that link therapeutic small molecules together, or link them with Eph-targeting peptides, proteins, and antibodies, have emerged as viable strategies for the creation of multi-modal therapeutics. While these next generation therapeutics hold great promise, eventually they must be adequately vetted to ensure safety and efficacy before heading to the clinic. As a result, significant delays can be anticipated before the promise of many of these exciting strategies is fully realized.

The past decade of small molecule-based Eph targeting has been fruitful, with the identification of numerous small molecules and peptides that enable specific targeting of Eph receptors (see summary, Table 1) [17,114]. In addition, Ephrin-targeting peptide sequences have been identified and further modified to improve specificity and bioavailability. In a recent report, the crystal structures of engineered variants of the EphA2-targeting peptide YSA with the EphA2 ligand binding domain were reported, providing an experimental framework for improving the affinity of EphA2-targeting peptides from the micromolar domain to the nanomolar domain and beyond [115]. Moreover, a new pH-dependent membrane peptide (TYPE7) was created based on the transmembrane domain sequence of EphA2 [116]. This peptide undergoes pH-dependent membrane insertion, and has an EphA2-targeting profile like that of ephrin-A1. Novel peptides with an affinity for the EphA2 receptor (peptide 135 H11, an optimized variant of YSA-derived peptide 123B9 [117]) and the EphA4 receptor (peptide 123C4, [118]) have also been investigated as potential pancreatic cancer treatments and therapeutics for amylotrophic lateral sclerosis (ALS). Small molecules have been extensively investigated for Eph receptor-mediated therapeutic potential, via screens of libraries of kinase inhibitors [119] and ATP-competitive inhibitors [120]. Structure–activity relationships and bioavailability studies for novel antagonists of EphA1 and EphB1 have also been reported [121]. The small molecule UniPR500, an EphA5 inhibitor, has undergone evaluation for its ability to improve glucose homeostasis in diabetic mice [122]. A novel EphB2-targeting molecule, HMQ-T-B10, was investigated as a potential anti-tumor therapeutic in hepatocellular carcinoma [123]. The EphB2-targeting properties of berberine have also been explored for anti-cancer activity, both alone [124] and in combination with the novel small molecule TPD7 [125]. Of further note, numerous peptide/protein/antibody–small molecule conjugates have been investigated for their ability to simultaneously target (peptide/protein/antibody) and treat (small molecule) conditions mediated by Eph receptors [126]. For example, the fusion of EphA2-targeting peptides and paclitaxel has been an active area of investigation [127,128,129,130], in addition to a fusion of gemcitabine and an EphA2 peptide targeted to pancreatic cancer [131]. EphA2-targeted antibody–drug conjugates have also been evaluated for their ability to target and treat tumors [132], and anti-EphA4 antibodies conjugated to the DNA-damaging reagent calicheamicin have been used in vivo to target tumor-initiating cells in triple-negative breast cancer and ovarian cancer [133]. Recently, Eph-targeted therapeutic strategies have also embraced therapeutic proteins, as radiosensitizers in cancer treatment (sEphB4-HAS, [134]) and as Eph-targeted cytotoxins (eA5-PE38QQR, [135]). A bispecific antibody, targeting both EphA2 and EphA3, has been explored as a potential therapy for glioblastoma [136], while antibodies targeted to EphB4 have been investigated for anti-cancer activity [137]. Finally, numerous RNA-based strategies have been explored for targeting Eph receptors [138]. Recent efforts include the synergistic knockdown of EphB2 with siRNA coupled with radiotherapy [139], and the targeting of EphA2 with microRNAs [140,141].

## 5. Targeted Delivery Strategies for Ephrins

Targeted smart drug delivery strategies to provide specific, rapid, non-invasive, biocompatible and robust Eph/ephrin RTK signaling benefits to the cell/tissue or region of interest, while avoiding off-target and/or toxic effects, are actively being pursued. For example, liposomes, microspheres, gels and biodegradable polymer nanoparticles are being engineered to replace conventional formulations and administration methods, and have demonstrated early success, particularly in cancer therapy (see summary Table 2) [142,143,144,145].

Eph-targeted liposomes have been formulated for the delivery of multiple therapeutic cargoes, including small molecule therapeutics, miRNAs, siRNAs and gene therapies. Efforts to target liposomes to Eph receptors include the EphA2-targeted delivery of doxorubicin and siRNA from nanoliposomes [146], EphA10-targeted delivery of doxorubicin and siRNA from lipoplexes [147], and EphA2-targeted delivery of doxorubicin from liposomes [148]. Nanoliposome formulations targeted to EphA2 have also been used to deliver a therapeutic microRNA (let-7a) to lung cancer cells, significantly increasing the delivery over treatment with miRNA alone [149]. Recently, an EphA2-targeting liposome encapsulating a taxane-prodrug was developed and tested for efficacy in lung and breast cancer models [150], with the targeted liposome formulation possessing higher efficacy than delivery of the drug alone. Nanoliposomes have also been linked to Eph-targeted single chain variable fragment (scFv) antibodies as a potential mode of therapeutic delivery. For example, targeted delivery of a liposomal cytotoxic nanoparticle was enabled by coupling with an engineered anti-EphA2 scFv [151,152]. In recent years, polymeric scaffolds have also been exploited for numerous Eph-targeting applications. These include the use of antibody-labeled PEG-ylated polymers for targeting the EphA2 receptor [153], and peptide-labeled cationic polymers targeted to EphA2 and EphA4 for therapeutic gene delivery [154]. EphB4-targeting biopolymer scaffolds have also been developed for the induction of the differentiation of neural stem cells [155]. Interestingly, DNA has been functionalized with the EphA2-targeting peptide SWL to create DNA-SWL nanostructures [156]. These nanostructures were shown to target EphA2-expressing prostate cancer cells.

Numerous nanoparticle-based formulations have been designed for the targeting of ephrins and Eph signaling pathways. For example, peptide-coated nanoparticles have also been targeted to EphA2 for the delivery of anti-cancer compounds [157]. Recently, a multi-faceted, EphA2-targeted, black phosphorous-based nanosystem was reported [158]. This nanoformulation uses the YSA peptide for EphA2-targeting and carries two therapeutic moieties: siRNA (interleukin-1alpha silencing) and the small molecule paclitaxel. In another report, the YSA peptide was used to functionalize gold nanorods, resulting in anti-proliferative activity on prostate cancer cells [159]. Eph-targeted nanoparticle formulations can also be the basis for advanced theranostic applications. For example, erythrocyte-derived nanoparticles were targeted to ephrin-B2 ligands via functionalization with the ligand binding domain of EphB1 [160]. Inclusion of a NIR-active dye (near infrared) in the nanoformulation makes them a promising platform for long wavelength phototherapies targeted to diseased cell populations. Ephrin-targeted nanoparticles have been developed to target glioblastoma. In one recent study, antibody-labelled polylactide-co-glycolide (PLGA) nanoparticles, targeting EphA3, were used for delivery of temozolomide via a nose-to-brain delivery route [161]. PLGA nanoparticles have also been targeted to the EphA2 receptor by functionalization with the YSA peptide, and their uptake in bleomycin-damaged lungs and cells investigated as a potential route for therapeutic delivery [162]. Nanoparticles have also been used to deliver novel recombinant proteins. For example, chitosan-coated nanoparticles bearing the recombinant protein Ephrin-A1-PE38/GM-CF were demonstrated to have anti-tumor activity in rats [163].

Finally, exosomes have been identified as a potentially powerful strategy for using Eph/ephrins to target select populations of cells and neurons [164]. Exosomes displaying membrane-bound Eph receptors were recently demonstrated to facilitate cell-contact-independent communication between cells [165,166]. As exosomes also have the capability of transporting therapeutic genes, proteins, and small molecules, they comprise another important class of Ephrin-targeted drug delivery that warrants further investigation. Exosomes have also been shown to promote angiogenesis through transport of Eph receptors [167]. Targeting or harnessing the intrinsic properties of EphR-laden exosomes may present a novel therapeutic route for inhibiting angiogenesis in cancer [168]. Exosomes may also transmit chemoresistance via delivery of EphA2 [169]. In this regard, targeting exosomal EphR could be a novel route for inhibiting chemoresistance. Finally, exosomes from cancer cells are known to transfer miRNAs to other cells within tumors [170]. This suggests that exosomes could perhaps be hijacked to deliver therapeutic small RNAs as well [171]. For example, exosomes from human cardiac progenitor cells have been shown to improve cardiac function and reduce apoptosis after myocardial infarction, through the transfer of microRNAs in infarcted hearts [172].

## 6. Conclusions

The intricate, multifaceted, and pervasive nature of the Eph/ephrin RTK system is being meticulously studied using an assortment of sophisticated molecular, cellular and in vivo techniques. These highly conserved proteins are a new and rapidly growing area of research as they influence a range of cellular behaviors and biological processes, thus possessing enormous translational potential for the treatment of human diseases. Currently, there are several clinical trials at different stages of recruitment/completion in a broad range of cancers. Many unanswered questions remain concerning the relative influence of these signaling pathways, not only on each other but on other mediators within the cell, as well as their interactions with neighboring cells, and the timing and cumulative/synergistic consequences of these events present a daunting challenge for discovering solutions. However, this versatility also holds great promise for the future of personalized medicine.

## Figures and Tables

**Figure 1 pharmaceuticals-13-00112-f001:**
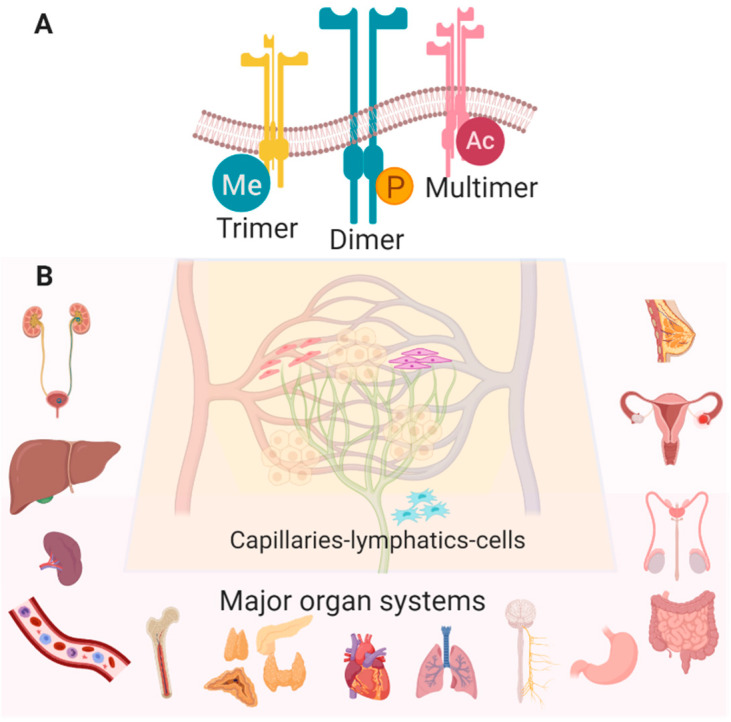
(**A**) Many configurations and post-translational modifications of Eph/ephrin RTKs have been observed (Me—methylation, P—phosphorylation, Ac—acetylation). (**B**) Eph/ephrin RTKs are widely expressed in various cell types in most healthy tissues with the following exceptions: EphA8 is detected only in spleen, brain and testes, EphA10 is in testes only, and ephrinA2 is absent in lung, spleen, testes, and bone marrow (Created with BioRender.com).

**Figure 2 pharmaceuticals-13-00112-f002:**
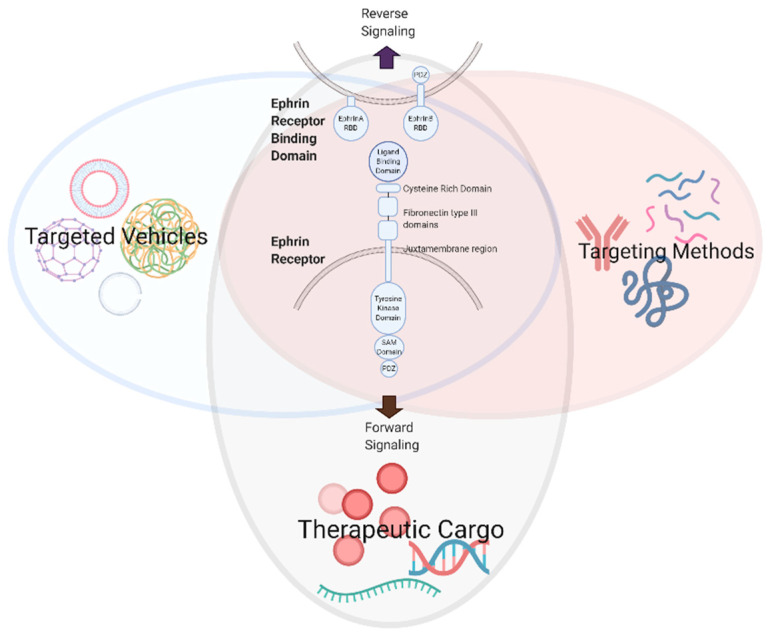
Development of therapeutic modalities to modulate Eph/ephrin signaling (created with BioRender.com).

**Table 1 pharmaceuticals-13-00112-t001:** Peptides, proteins, and small molecules developed to target Eph Receptors.

Target Receptor	Agent Type	Agent Name	Sequence/Description	Reference
EphA2	Peptide	YSA	YSAYPDSVPMMS	[115]
	Peptide	YSA-GSGSK-bio	YSAYPDSVPMMSGSGSK-bio	[115]
	Peptide	SWL	SWLAYPGAVSYR	[115]
	Peptide	WLAam	WLAYPDSVPMam	[115]
	Peptide	βA-WLA-YRPK-bio	βAWLAYPDSVPYRPK-bio	[115]
	Peptide	βA-WLA-YSK-bio	βAWLAYPDSVPYSK-bio	[115]
	Peptide	TYPE7	EFQTLSPEGSGNLAVIGGVAVGVVLELVLAGVEFFIEEEEE	[116]
	Peptide	123B9	(4-F,3-ClPhOCH_2_CO)SAYPDSVP(Nle) (hS)S-CONH_2_	[117]
	Peptide	135H11	XLA(4MeTyr)PDA V(Hyp)(4ClPhe)RP-CONH2 X = 3-methyl-6,7-dimethoxy-benzofuranoic acid	[117]
	Peptide-small molecule	(123B9)_2_–L2–PTX	Dimeric 123B9 conjugated to paclitaxel	[127,128,129,130]
	Peptide-small molecule	123B9-L2-Gem YNH/YDH-L2-Gem	123B9 peptide conjugated to gemcitabine YNH or YDH peptide conjugated to gemcitabine	[131]
	Antibody-drug	3B10-ADC 1C1-ADC	anti-EphA2 monoclonal antibodies fused to tubulysin variant AZ13599185	[132]
	Small molecule	UniPR139, UniPR502		[121]
	microRNA	miRNA-302B; miRNA-26B		[140,141]
EphA4	Peptide	123C4		[118]
	Antibody-drug	PF-06647263	hE22 monoclonal antibody fused to calicheamicin	[133]
EphA5	Small molecule	UniPR500		[122]
EphB2	Small molecule	HMQ-T-B10		[123]
	Small molecule	berberine		[124]
	Small molecule	TPD7		[125]
	siRNA		EphB2 knockdown + radiation	[139]
EphB4	Protein	sEphB4-HAS	EphB4 receptor fragment fused to human serum albumin	[134]
	Antibody	H200 pAb	Polyclonal antibody raised against 200 aa extracellular region of EphB4	[137]
Multiple targets	Protein fusion	eA5-PE38QQR	EphR ligand eA5 fused to truncated form of Pseudomonas aeruginosa exotoxin A	[135]
	Antibody	EPHA2/A3 BsAb	Novel bispecific antibody targeting EphA2 and EphA3	[136]

**Table 2 pharmaceuticals-13-00112-t002:** Engineered delivery systems designed to improve target specificity.

Target Receptor	Agent Type	Description	Reference
EphA2	Liposome	YSA-liposomes for co-delivery of doxorubicin and JIP1 siRNA	[146]
EphA10	Liposome	EphA10 antibody lipoplex for co-delivery of doxorubicin and MDR1-siRNA	[147]
EphA2	Liposome	YSA-liposomes for delivery of doxorubicin	[148]
EphA2	Liposome	Eph1A-liposomes for delivery of let-7a miRNA	[149]
EphA2	Liposome	Delivery of paclitaxel and docetaxel prodrugs	[150]
EphA2	Lipsome	scFv-liposome for delivery of cytotoxin	[151,152]
EphA2	Polymer	scFV 4B3-pegylated hyperbranched polymer	[153]
Multiple targets	Polymer	CHVLWSTRC-peptide labeled cationic polymer delivers therapeutic sRAE-1γ plasmid via EphA2 and EphA4 receptors	[154]
EphB4	Polymer	Biopolymer functionalized with ectodomain of ephrinB2	[155]
EphA2	DNA	Ephrin-A1 decorated DNA nanostructure	[156]
EphA2	Nanoparticle	Pegylated EphA2 peptide coated nanoparticles	[157]
EphA2	Nanoparticle	YSA-nanoparticle for co-delivery of ILsi RNA and paclitaxel	[158]
EphA2	Nanorod	YSA-gold nanorods	[159]
EphB1	Nanoparticle	EphB1 ligand binding domain-erythrocyte nanoparticles for delivery of phototherapy	[160]
EphA3	Nanoparticle	EphA3 antibody-nanoparticles for delivery of temozolomide	[161]
EphA2	Nanoparticle	YSA-polymeric nanoparticles	[162]
EphA2	Nanoparticle	Chitosan-coated Ephrin-A1-PE38/GM-CSF nanoparticles	[163]

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
