# Peer review of "Harnessing the Power of Eph/ephrin Biosemiotics for Theranostic Applications"

_pharmaceuticals, 2020, doi:10.3390/ph13060112_

Round 1

Reviewer 1 Report

Dear authors

the review is too generic  and there are already  published reviews  (also reported in the references) that argue the topic in generic way describing several aspects or applications. So I cannot accept the work in this way, for this reason  I suggest to  choose one specific new topic correlated to Eph/Eph system on which build up  all the review. Further it's necessary to  change the title using clear and direct words and not unnecessaritly sophisticated  ( see, as examples, reviews with the same topic)

Author Response

We thank the reviewer for the time and effort in reading and commenting on this manuscript. This review is intended to provide an overview of recent discoveries of the role that ephrin/Eph signaling plays in health and disease, the possibilities for exploitation in treatment of several diseases in most tissues, and the newly developed approaches being explored to provide directed therapeutics. While we appreciate the reviewer’s opinion, we chose this approach for this specific call as a means to highlight the diversity that is anticipated to be included in this compilation. The title contains scientific terminology that is purposely directed at this audience and it was formulated to draw attention to the fact that this field is rapidly gaining ground on all fronts as addressed in this overview.

Reviewer 2 Report

This review article discusses current progress, knowledge gap and potential avenues for further investigation on Eph/ephrin system. Overall the manuscript is well written. I have a couple of minor suggestions as follows -

  1. The authors added all the references at the end of the introduction section (1-17) and for line 68 (1-17, 31-37). I think it's too many references all together for a whole section. please break down the references accordingly or add specific references for the statements.
  2. I would prefer if the authors can provide a schematic diagram for the Eph/ephrin signaling dynamics, it'll be more understandable for readers instead of going through the extensive texts.

Author Response

We thank the reviewer for the time invested in reading this manuscript and providing these constructive comments. We have addressed the first point regarding the citations concentrated at the end of the introduction and in line 68.  Regarding the schematic, our intention was not to discuss the specifics of the signaling cascades but rather to address the development of a range of therapeutic strategies to specifically target signaling to ameliorate disease processes. We have created a graphic (Figure 1) that hopefully captures this concept and satisfies this request.

Reviewer 3 Report

The authors well describe recent knowledge about the Eph/ephrin study mainly focused on cancer in this review. The manuscript has a potential to be a good text book for broad scientific readers. However, the reviewer feels that the manuscript needs minor corrections. Especially, misspellings are too many. Therefore, an extensive proofreading is required.

Concerns

1) In line 24 on page 1 and line 79 on page 2, what is "Eph/ephrin receptor"? This definition is not general. Ephrin is not a receptor tyrosine kinase. Should be generalized and corrected.

2) Figures would be helpful for readers especially in paragraph 1 of Eph/ephrin is a ubiquitous therapeutic target. Figures should be added.

3) In line 63 on page 2, RTKS should be corrected.

4) In figure 1 legend and line 88 on page 2, ephrin/Eph should be corrected.

5) Ephrin should be connected with a hyphen in front of subclass and number represented as ephrin-A1.

6) In  line 179 on page 4, EphA1-R should be corrected. EphA1 is a receptor. Therefore, -R should be removed.

7) In line 184 on page 4, Ca2+ should be corrected.

8) In line 188-190 on page 4, a reference is required.

9) In line 192 on page 4, explanation of EphA4-Fc is required. Fc-fused ephrin-A1 is not existed in physiological and pathological settings in nature.

10) In line 226 on page 5, what is EphrB2? Should be corrected.

11) In line 271 on page 7, E of "Ephrins" should be expressed as e.

12) In line 272 on page 7, EphR should be expressed as Eph. Eph means a receptor.

13) In line 283 on page 7, what is Eph3? Should be corrected.

Author Response

We thank the reviewer for the valuable and specific feedback to improve the quality and comprehensibility of this review.

The following responses address the specific concerns raised:

1) In line 24 on page 1 and line 79 on page 2, what is "Eph/ephrin receptor"? This definition is not general. Ephrin is not a receptor tyrosine kinase. Should be generalized and corrected.

Response: We apologize for this oversight and have modified it appropriately.

2) Figures would be helpful for readers especially in paragraph 1 of Eph/ephrin is a ubiquitous therapeutic target. Figures should be added.

Response We have created a new figure (1) to represent the complex concepts in the ubiquitous target paragraph. We welcome specific ideas and feedback to improve/modify.

3) In line 63 on page 2, RTKS should be corrected.

Response: This error has replaced S with s.

4) In figure 1 legend and line 88 on page 2, ephrin/Eph should be corrected.

Response: This mistake has been fixed.

5) Ephrin should be connected with a hyphen in front of subclass and number represented as ephrin-A1.

Response: Thank you for drawing this to our attention. We have placed the missing hyphen

6) In  line 179 on page 4, EphA1-R should be corrected. EphA1 is a receptor. Therefore, -R should be removed.

Response: The -R has been deleted.

7) In line 184 on page 4, Ca2+ should be corrected.

Response: The + has been moved to the right of the 2.

8) In line 188-190 on page 4, a reference is required.

Response: We apologize for this omission and have added this reference.

9) In line 192 on page 4, explanation of EphA4-Fc is required. Fc-fused ephrin-A1 is not existed in physiological and pathological settings in nature.

Response: The definition of Fc has been included in parentheses.

10) In line 226 on page 5, what is EphrB2? Should be corrected.

Response: The r has been deleted.

11) In line 271 on page 7, E of "Ephrins" should be expressed as e.

Response: The E has been replaced with e.

12) In line 272 on page 7, EphR should be expressed as Eph. Eph means a receptor.

Response: The R has been deleted.

13) In line 283 on page 7, what is Eph3? Should be corrected.

Response: The omitted A has been added.

Reviewer 4 Report

The manuscript entitled "Harnessing the Power of Ephrin/Eph Biosemiotics for Theranostic Applications" has been reviewed carefully for the publication. Author s have performed the extensive literature review and presented a well-compiled review about the "Eph/ephrin system" topic. The manuscript is well written and surely will add new insight into the topic to the existing literature. The manuscript can be accepted for publication in its current form after minor grammatical editing.

Author Response

We thank this reviewer for positive feedback and the constructive comments. We have done our best to edit the grammatical and spelling errors throughout.

Reviewer 5 Report

It is a pretty clear and very-well organized structure for overviewing such a complicated "Eph/ephrin system".

For the "signaling dynamics" part, it would be more straightforward to draw some schematic diagrams to cover the role of Eph/ephrin signaling, which was discussed in the following sections. Would it be possible to reveal some kind of crosstalk between the different "roles" that have been mentioned? Or some kind of crosstalk with other classic signalings?

Great job!

Author Response

We thank this reviewer for the time and effort in evaluating our manuscript and are very excited that it was found to be readable and impactful. We have created a new figure to capture the concepts discussed pertaining to the complex signaling dynamics and hope this satisfies this request. We welcome specific ideas and feedback to modify/improve.